

# Hydrogen and carbon isotope fractionation factors of aerobic methane oxidation in deep-sea water

Shinsuke Kawagucci[1][2], Yohei Matsui[3][4], Akiko Makabe[1], Tatsuhiro Fukuba[5], Yuji Onishi[1][6], Takuro Nunoura[7], Taichi Yokokawa[1]

[1] Super-cutting-edge Grand and Advanced Research (SUGAR) Program, Institute for Extra-cutting-edge Science and Technology Avant-garde Research (X-star), Japan Agency for Marine-Earth Science and Technology (JAMSTEC), Yokosuka 237-0061, Japan
[2] Institute of Geochemistry and Petrology, ETH Zürich, Zürich 8092, Switzerland
[3] Advanced Science-Technology Research (ASTER) Program, X-star, JAMSTEC, Yokosuka 237-0061, Japan
[4] Department of Engineering Mechanics and Energy, University of Tsukuba, Tsukuba 305-0006, Japan
[5] Institute for Marine-Earth Exploration and Engineering (MarE3), JAMSTEC, Yokosuka 237-0061, Japan
[6] The Center for Ecological Research, Kyoto University, Otsu 520-2113, Japan
[7] Research Institute for Marine Resources Utilization (MRU), JAMSTEC, Yokosuka 237-0061, Japan
Correspondence to: Shinsuke Kawagucci (kawagucci@jamstec.go.jp)
& Taichi Yokokawa (taichi.yokokawa@jamstec.go.jp)

**Abstract. (248 words)** Isotope fractionation factors associated with various biogeochemical processes are important in ensuring the practicality of isotope tracers in biogesciences at large. Methane is a key component of the subsurface biosphere and a notable greenhouse gas, making the accurate evaluation of methane cycles, including microbial
methanotrophy, imperative. Although the isotope fractionation factors associated with methanotrophy been examined under various conditions, the dual-isotope fractionation factors of aerobic methanotrophy in oxic seawater column remain unclear. Here, we investigated hydrogen and carbon isotope ratios of methane as well as the relevant biogeochemical parameters and microbial community compositions in hydrothermal plumes in the Okinawa Trough. Methanotrophs were found to be abundant in plumes above the Hatoma Knoll vent site, and we succeeded in simultaneously determining hydrogen and
carbon isotope fractionation factors associated with aerobic oxidation of methane ($\varepsilon^H$ = 49.4±5.0‰, $\varepsilon^C$ = 5.2±0.4‰) – the former being the first of its kind ever reported. This $\varepsilon^H$ value is comparable with reported values from terrestrial ecosystems but clearly lower than those from aerobic and anaerobic methanotroph enrichment cultures, as well as incubations of methanotrophic isolates. The covariation factor between $\delta^{13}C_{CH4}$ and $\delta D_{CH4}$, $\Lambda$ (9.4/8.8 determined using two different methods), was consistent with those from methanotrophic isolate incubations. These values determined herein are valuable
for understanding dynamics of methane cycling in the marine realm, and future applications of the approach used herein to other habitats with methanotrophic activity will help reveal whether the small $\varepsilon^H$ value observed herein is a ubiquitous feature across all marine systems.



## 1 Introduction

Stable isotope ratios have been widely used for tracing biogeochemical cycles and microbial activities [e.g. Ohkouchi et al., 2010; Sharp, 2017]. To ensure the practicality of isotope tracers in biogeoscience, it is important to understand the isotope fractionation factors associated with expected processes in the environment of interest. The isotope fractionation factor associated with microbial metabolism, a series of multi-step enzyme-catalyzed reactions, is known to fluctuate due to physiological responses to environmental conditions such as temperature and substrate availability [e.g. Valentine et al.

2004]. As the complex biogeochemistry of a natural system cannot be practically rebuilt in full in either laboratory or models, determination of the fractionation factor in the natural environment is most appropriately done via observation.

Measuring multiple-isotope-ratios of a compound is a powerful tool to trace a specific process. Currently available multi-isotope-ratio approaches include mass-independent fractionations [e.g. Farquhar et al., 2000; Michalski et al., 2003],

position-specific differences [Blair et al., 1987; Yoshida & Toyoda, 2000], multiple substitution 'clumped' isotopes [Affek and Eiler, 2006; Ghosh et al., 2006], and multi-element isotope ratios of a compound [e.g. Vogt et al., 2016]. Among these, the two-dimensional analysis of dual-element isotope ratios in compounds, such as $\delta^{15}N$ and $\delta^{18}O$ of nitrite [Casciotti, 2016] and $\delta^{13}C$ and $\delta^{37}Cl$ values of methyl chloride [Konno et al., 2015], is the most conventional method used to date. As isotope effects associated with molecular physical dynamics, such as diffusion, occur to the same extent for each element in a

compound, the two-dimensional analysis is useful in distinguishing the processes producing and consuming the compound.

Methane ($CH_4$) is a representative compound of the subsurface environment [e.g. Ijiri et al., 2018] and is also a notable atmospheric compound due to its strong greenhouse effect [e.g. Saunois et al., 2020]. As such, the accurate evaluation of the marine $CH_4$ cycle is a matter of urgency [Reeburgh, 2007; Dean et al., 2018]. Two-dimensional analysis with carbon and

hydrogen isotope ratios has been applied to trace the origins and behaviors of $CH_4$ [Whiticar et al. 1986; Sugimoto and Wada, 1995]. Carbon and hydrogen isotope fractionation factors associated with the microbial consumption of $CH_4$, methanotrophy, have been determined through incubation of isolates [Feisthauer et al., 2011], incubation of enrichment cultures [Coleman et al., 1981; Kinnaman et al., 2007; Holler et al., 2009; Rasigraf et al., 2012], and observations in the natural environment [Snover and Quay, 2000; Kessler et al., 2006]. While wide variations have been observed in each isotope fractionation, with

a factor between 2-36‰ for carbon and 36-320‰ for hydrogen, the ratio between carbon and hydrogen fractionation factors falls into a narrow range of 6-15 regardless of magnitudes of the fractionations [Feisthauer et al., 2011; Rasigraf et al., 2012 and references therein]. The dual-isotope fractionation factors associated with methanotrophy have been examined under a variety of environmental and physiological properties such as aerobic/anaerobic, terrestrial/marine, temperatures, and forms of key enzyme methane monooxygenase [reviewed by Feisthauer et al., 2011; Rasigraf et al., 2012]. Despite these efforts,

the dual-isotope fractionation factors of methanotrophy in oxic seawater column have never been determined while only the carbon isotope fractionation factors of aerobic methanotrophy in oxic seawater column have been determined to date through



observations of CH₄-rich hydrothermal plumes [Tsunogai et al., 2000; Gharib et al., 2005; Gamo et al., 2010; Kawagucci et al., 2010].

Here, we studied both hydrogen and carbon isotope ratios of CH₄ as well as the relevant biogeochemical processes and microbial community compositions in plumes above two hydrothermal vents in the Okinawa Trough, Hatoma Knoll site and Daisan-Kume Knoll ANA site, in order to simultaneously determine hydrogen and carbon isotope fractionation factors associated with aerobic oxidation of CH₄ in the marine environment. High-resolution vertical samplings inside and above seamount craters, at the base of which the vents are located, allowed us to capture gradual biogeochemical alteration of the

plumes. Since vigorous venting of fluids with high CH₄ concentration (>10 mM) with CH₄-bearing boiling-derived bubbles at Hatoma Knoll [Toki et al., 2016] is contrastive with relatively weak discharge of fluids with lower CH₄ concentration (<1mM) at ANA site [Makabe et al., 2016], our initial intention was to reveal critical mechanisms controlling the fractionation factors by comparing them. We successfully determined at Hatoma site the first hydrogen isotope fractionation factor of aerobic methanotrophy in seawater column; along with carbon isotope fractionation factors consistent with those

reported to date [Tsunogai et al., 2000; Gharib et al., 2005; Gamo et al., 2010; Kawagucci et al., 2010].

## 2 Sampling and Analysis

### 2.1 Sampling

Seawater samples were collected during R/V Mirai cruise MR17-03C in 2017 [Brisbin et al., 2020] above two active deep-

sea hydrothermal vent sites (Figure 1), including Hatoma Knoll site (depth: 1531 m)[Toki et al. 2016] and Daisan-Kume Knoll ANA site (depth: 1079 m)[Makabe et al., 2016; Uyeno et al., 2020]. Hatoma Knoll is characterized by vigorous venting of fluids ≦323 oC with boiling-derived bubbles. The galatheoid squat lobster Shinkaia crosnieri, exhibiting ectosymbiosis with a microbial community of methanotrophs and thiotrophs [Watsuji et al. 2014], is dominant and densely distributed across the hydrothermally active area of Hatoma Knoll. Discovered in 2015 near Kumejima Island, ANA takes its

name from an acronym of "Acoustic anomaly from Narrow pit Areas on caldera floor" [Nakamura et al., 2015], with a double meaning where 'ana' means 'hole' in Japanese. The maximum measured vent fluid temperature of ANA site was 229 oC. The total vent fluid flux of ANA site appeared to be more than an order of magnitude lower than those of Hatoma Knoll [unpublished]. Active venting at both sites are located on the bottom of caldera at the top of knolls, with the surrounding caldera walls being approximately 200 m high (Figure 1).


High-resolution vertical sampling with 23 seawater samples collected within 400 m altitude from the seafloor were conducted just above the center of activity at both vent sites (Hatoma Knoll: 24°51.426'N – 123°50.328'E –1531m, ANA site: 26°17.490'N – 126°28.158'E – 1079 m) using a Conductivity Temperature Depth (CTD) profiler with Carousel



Multiple Sampling (CMS) system. The CTD-CMS system deployed during the MR17-03C cruise consisted of a CTD sensor
(SBE911plus, Sea-bird Scientific), a SBE32 Carousel water sampler (Sea-bird Scientific) for 36 Niskin-X bottles, a
dissolved oxygen (DO) sensor (RINKO), and a turbidity meter (Seapoint Turbidity Meter, Sea-bird Scientific). Vertical
profiles of temperature were drawn as difference from the bottom temperature (Figure 2a), which was 3.79 at Hatoma Knoll
and 4.45 at ANA site.

### 2.2 Analytical procedure

Concentrations and dual isotope ratios of $CH_4$ and nitrous oxide ($N_2O$) in the seawater samples were measured by a custom-
made purge-and-trap system with a MAT253 isotope-ratio mass spectrometer (IRMS), following previously published
methods [Hirota et al. 2010] and its modifications [Okumura et al., 2016; Kawagucci et al., 2018]. Immediately after
recovery of the CTD-CMS system on deck, each seawater sample in the Niskin bottle was subsampled into a 120 mL glass
vial, sealed with a butyl rubber septum and aluminum cap after the addition of 0.2 mL mercury chloride-saturated solution,
and stored at 4 °C until shore-based measurement at laboratories in Japan Agency for Marine-Earth Science and Technology
(JAMSTEC), Yokosuka.

Each seawater sample was transferred by helium stream (rate: 100 mL/min) into a 250 mL purge bottle to extract dissolved
gases by helium bubbling and magnetic stirring. $CH_4$ and $N_2O$ in the stripped gas were purified by passing through a
stainless-steel tubing coil trap held at −110 °C (ethanol/liquid-$N_2$ bath) as well as a chemical trap filled with magnesium
perchlorate ($Mg(ClO_4)_2$; Merck KGaA) and Ascarite II (sodium-hydroxide-coated silica; Thomas Scientific) followed by a
stainless-steel tubing trap filled with HayeSep-D porous polymer (60/80 mesh, Hayes Separations Inc.) held at −130 °C
(ethanol/liquid-$N_2$ bath). Then, the $CH_4$ and $N_2O$ on the HayeSep-D trap were released into another helium stream (rate: 1.0
mL/min) and again condensed on a capillary trap made of PoraPLOT Q (20 cm long, 0.32 mm i.d.) held at −196 °C (liquid-
$N_2$ bath) for cryofocus, and finally released at room temperature.

After the complete separation of $CH_4$ and $N_2O$ from the other molecules using a GS Carbon-PLOT capillary column (30 m
long, 0.32 mm i.d.) at 40 °C, the effluent $CH_4$ was put through a 960 °C combustion unit (Thermo Fisher Scientific) to
convert into CO2 prior to introduction into MAT253 via an open-split interface (GCC III, Thermo Fisher Scientific). On the
other hand, $N_2O$ was introduced into GCC III and IRMS without conversion. Carbon, nitrogen, and oxygen isotope ratios
were obtained through simultaneous monitoring of $CO_2^+$ and $N_2O^+$ isotopologues at m/z = 44, 45, and 46 by Faraday cups
with 10 times-higher resistors, for increased sensitivity compared to commercially-available general settings [Kawagucci et
al., 2018]. For determination of hydrogen isotope ratio of $CH_4$, another vial of the same sample was used, the analytical
procedure being almost identical to the above except a 1440 °C pyrolysis unit (Thermo Fisher Scientific, Massachusetts,
USA) was applied for conversion into $H_2$, m/z = 2 and 3 [Okumura et al., 2016]. Isotope ratios are represented by
conventional δ notation and presented in permil scale. Errors for the analyses of deep-sea water conducted during the present



study were estimated from repeated analyses of a sample, and were within 10% for $N_2O$ concentration, 0.2‰ for $\delta^{15}N_{N2O}$, 0.5‰ for $\delta^{18}O_{N2O}$, 20% for $CH_4$ concentration, 0.3‰ for $\delta^{13}C_{CH4}$, and 5‰ for $\delta D_{CH4}$.

Isotope fractionation factor associated with kinetic effect, ε (‰), can be expressed by the following equation:

$$\varepsilon = (^{h}k/^{l}k) - 1 \qquad (1),$$

where k is reaction kinetics while $(^{h}k/^{l}k)$ is the kinetic isotope effect and superscripts h and l represent the heavier isotopologue (e.g. $^{13}C$) and the lighter one (e.g. $^{12}C$), respectively. Observations of fractional yield and isotope composition of the reactant in a closed system can be used to determine the fractionation factor using the following equation [Mariotti et

al., 1981]:

$$\varepsilon = [\delta_{rx} - \delta_{r0}]/\ln(1 - f) \qquad (2),$$

where the subscripts designate initial reactant (r0) and reactant remaining in a sample (rx), and f represents the fraction of the reactant consumed after the reaction. When δrx is plotted as a function of ln(1-f), the ε value is given by the slope of the line. A factor expressing covariation between $\delta^{13}C_{CH4}$ and $\delta D_{CH4}$, Λ, is defined [Elsner et al. 2010; Feisthauer et al. 2011] by the

following equation:

$$\Lambda = \varepsilon^{H}/\varepsilon^{C} \qquad (3).$$

Assuming variations in the isotope ratios of $CH_4$ in the environment are attributable solely to $CH_4$ oxidation, Λ can be calculated also using another equation without the parameter f from discriminations of isotope ratios observed [Tsunogai et al., 2020], as follows:

$$\Lambda = (\delta D_{rx} - \delta D_{r0})/(\delta^{13}C_{rx} - \delta^{13}C_{r0}) \qquad (4).$$

The manganese concentration was measured in a laboratory of KaiyoKeisoku Co. Ltd. (Prof. Kei Okamura) using 100 mL of seawater without filtration using the luminol-$H_2O_2$ chemiluminescence detection method [Ishibashi et al., 1997; Kawagucci et al., 2018] calibrated with the international standard NASS-5. Ammonium concentrations were determined on-board the

research vessel with the conventional AutoAnalyzer method.

The concentration of particulate ATP (pATP) was determined by a luciferin-luciferase assay on-board the ship. Seawater samples collected using the CTD-CMS were transferred aseptically to clean plastic tubes, and an aliquot of this subsample was immediately filtrated through a satirized membrane filter unit (0.2 μm pore size). Concentrations of ATP in both

unfiltered and filtered aliquots were measured with a simplified quantification assay using the ATP assay kit (CheckLight HS, Kikkoman Biochemifa) without pre-concentration and extraction process. A 100 μL of cell lysis reagent was directly added to 100 μL of seawater sample for ATP extraction in a disposable test tube. After leaving the lysate for at least 30 minutes at room temperature to minimize the effect of temperature differences between the samples, 100 μL of the luciferin-luciferase mixture reagent was added to the lysate and the luminescence intensity was measured immediately using a desktop

photodetector (Luminescencer Octa AB-2270, ATTO). The ATP concentrations from unfiltered and filtered subsamples



were regarded as total ATP and dissolved ATP, respectively. The pATP concentration was calculated by subtracting the dissolved ATP concentration from the total ATP concentration.

Samples for measuring the microbial cell density were fixed with 0.5% (wt/v) glutaraldehyde (final concentration) in 2 mL
cryo-vials on-board and stored at -80 °C until further analysis. For cell density measurements, 200 μL of each sample were stained by SYBR Green I nucleic acid gel stain (ThermoFisher Scientific, ×5 of manufacture's stock) at room temperature for >10 min. The total microbial cell abundance in 100 μL of sample was determined using an Attune NxT Acoustic Focusing Flow Cytometer (ThermoFisher Scientific) by their signature in a plot of green fluorescence versus side scatter [Brussaard, 2004; Giorgio et al., 1996].


To obtain microbial community structure based on amplicon sequencing of 16 rRNA gene, 2–4 L of seawater samples were filtered with 0.2-μm-pore-size cellulose nitrate or acetate membrane filters. The filters were stored at −80°C until environmental DNA extraction, following previously published methods [Hirai et al., 2017]. The 16S SSU rRNA gene was amplified from the extracted DNA with the primer mixture of 530F and 907R, using the LA Taq polymerase with GC buffer
(Takara Bio) as described previously [Nunoura et al., 2012; Hiraoka et al., 2020]. The amplicon sequencing libraries were sequenced using an Illumina MiSeq high-throughput sequencing (2×300 paired-end platform) at JAMSTEC. The sequence data generated herein are publicly available in the DDBJ sequence read archive (DRA) under the BioProject PRJDB11835.

Raw paired-end reads were merged using PEAR v0.9.10 [Zhang et al. 2014], and primer sequences were removed using
Cutadapt v1.10 [Martin et al. 2011]. Low-quality (Q score <30 in over 3% of sequences) and short (<150 bp) reads were filtered out using a custom perl script. A total of 5,996,472 SSU rRNA gene sequences from 59 samples were analyzed using QIIME2 v 2019.4.0 pipeline (Bolyen et al. 2019). Unique amplicon sequence variants (ASVs) were generated using the DADA2 plugin wrapped in QIIME2 and chimeric sequences were removed [Callahan et al. 2016]. The taxa were assigned to the ASVs for 16S rRNA genes using the QIIME2 plugin feature-classifier classify-sklearn [Bokulich et al. 2018] against the
SILVA 138 database [Quast et al. 2013].

### 3 Results

#### 3.1 Hydrothermal plume signature

Vertical profiles of biogeochemical parameters drawn by altitude from the seafloor are presented in Figure 2 while all the
results are provided as Supplementary Table S1. Water columns showed simultaneous increases in turbidity, manganese, and CH₄, and demonstrated the presence of hydrothermal plumes above Hatoma Knoll and ANA site. Temperature changes (from the bottom water temperature) fluctuated at both sites along the vertical profile, with no clearly identifiable peaks. DO





profiles at both sites demonstrated sufficiently oxic seawater for aerobic metabolisms through the water column observed. The turbidity profile above Hatoma Knoll exhibited two vertically-broad turbid water masses, one near the seafloor and

another at 100-200 m altitude. At ANA station the turbidity profile displayed a single peak centered around 130 m altitude. The maximum turbidity seen above Hatoma Knoll site was an order of magnitude lower (0.6 FTU) than ANA site. Drastic change in temperature and decline of turbidity above 200 m altitude at Hatoma Knoll plume suggests the height of the caldera rim. Vertical patterns of manganese concentrations at each station were generally similar to turbidity. The maximum manganese concentrations observed were also lower at Hatoma Knoll (21 nM) than ANA site (119 nM).


Vertical distribution of $CH_4$ concentrations also showed similar patterns to those of turbidity and manganese, but differed from them in that the maximum $CH_4$ concentrations observed were clearly higher at Hatoma Knoll (935 nM) compared to ANA site (254 nM). The $CH_4/Mn$ ratios varied between 1 and 45 above Hatoma Knoll, but were roughly constant around 1.5 in the water column above ANA site (Figure 2f).

**3.2 Methane isotope composition**

Carbon and hydrogen isotope ratios of $CH_4$ were distinct between the two vent sites. Above Hatoma Knoll, vertical patterns of $CH_4$ concentrations and the $\delta^{13}C_{CH4}$ values were mirror images of each other, with the $\delta^{13}C_{CH4}$ values being the lowest (approximately -47‰) at 107 m altitude where the $CH_4$ concentrations were high, while the highest $\delta^{13}C_{CH4}$ value (-35.6‰) appeared at 77 m altitude where the $CH_4$ concentrations were depressed between the twin peaks. Vertical pattern of $\delta D_{CH4}$

values above Hatoma was similar to that of $\delta^{13}C_{CH4}$ value, peaking at 77 m altitude with +17‰. Above ANA site, however, the $\delta^{13}C_{CH4}$ and $\delta D_{CH4}$ values were almost constant across the depth gradient at -28‰ and -110‰, respectively.

The $^{13}C$-D diagram for $CH_4$ observed above Hatoma Knoll demonstrated a linear trend (Figure 3). Isotopic composition of the low-$\delta^{13}C_{CH4}$ root of the linear distribution is consistent with the $\delta^{13}C_{CH4}$ values of Hatoma vent fluids determined by

direct sampling (-54–-49‰)[Toki et al., 2016], and the $\delta D_{CH4}$ values determined from all known deep-sea vent fluids (approximately -120‰)[Proskurowski et al., 2006]. A least-square linear fitting for the dataset obtained above Hatoma Knoll yielded a slope of 8.8 (Figure 3), which corresponds to Λ according to Equation (4).

**3.3 Nitrogen species**

Ammonium were more enriched in water column above Hatoma Knoll compared to ANA site (Figure 2i). The vertical pattern of ammonium concentrations above Hatoma was similar to that of $CH_4$ concentrations. Regardless of the nitrogen lineage of ammonium, $N_2O$ concentrations and nitrogen and oxygen isotope ratios were constant in both Hatoma and ANA hydrothermal plumes at 28.8±0.7 nM and 29.1±1.4 nM, +8.7±0.1‰ and 8.5±0.2‰ ($\delta^{15}N_{N2O}$), and +55.1±0.3‰ and +54.3±0.6‰ ($\delta^{18}O_{N2O}$), respectively.




### 3.4 Microbiological characteristics

Total cell density (cells mL$^{-1}$) above Hatoma Knoll generally fell within a narrow range between 1.9-7.6×10$^4$ across the depth gradient (Figure 2m). In the cell counting using flow cytometry, the cytogram obtained exhibited two clusters, each of which represented the typical microbial cells and cells with typical fluorescence signals with significantly higher side-scatter

signals, named High side scatter population (HSS). The HSS is attributable to a big cell and/or cell aggregate. Only two samples between 200–230 m showed significant HSS cell abundances at Hatoma Knoll. At ANA site, total cell density below 250 m altitude were an order of magnitude higher (≤4.9×10$^5$ cell mL$^{-1}$) than that above 250 m (4.0×10$^4$ cell mL$^{-1}$). Between 40–160 m altitude, HSS cells occupied 20-58% of the total cells. The cell density in these samples is possibly underestimated due that a HSS signal possibly consists of cells as aggregate.


pATP concentrations above Hatoma Koll increased toward the deep and were nearly constant below 200 m (the height of the caldera rim) at an order of 10$^1$ pmol L$^{-1}$ (Figure 2n). Above ANA site, pATP concentrations were the highest between 33–140 m altitude at 3.0×10$^2$ pmol L$^{-1}$. pATP concentrations above 250 m altitude were comparable between the two vent sites at the 100 order.


Cellular ATP contents (ng-ATP cell$^{-1}$) below 10$^{-7}$ were observed above 300 m altitude above both Hatoma Knoll and ANA site (Figure 4). The level observed is consistent with values previously reported from open ocean water [e.g. Winn & Karl, 1986], suggesting that they represent background levels without any effect from hydrothermal input. Gradual increase of the cellular ATP content along with increased depth was found from the caldera rim depth (200 m altitude) to the seafloor at

Hatoma as well as across the depth gradient above ANA site. Cellular ATP contents above 10$^{-7}$ ng-ATP cell$^{-1}$, despite the low cell density (<10$^5$ cells mL$^{-1}$), were observed below 200 m altitude at Hatoma Knoll.

Microbial community analysis using amplicon sequence of 16S rRNA gene revealed distinct communities between Hatoma Knoll and ANA site (Figure 5). In water column from the seafloor to 200 m altitude above Hatoma site, more than a half of

the community consisted of members of SUP05 (composed of three ASVs: SUP05_1, _2 and _3), a well-known group of chemolithotrophic sulfur-oxidizing Gammaproteobacteria frequently detected around hydrothermal vents and within hydrothermal plumes [Dick et al., 2013]. Approximately 10% of the community was occupied by the aerobic methanotrophic Gammaproteobacteria family Methylococcaceae (composed of two ASVs: Methylococcaceae_1 and _2) and >5% by the ammonium-oxidizing archaeal family Nitrosopumilaceae (composed of two ASVs: Nitrosopumilaceae_1 and

_2). Occupancies of SUP05 decreased with increasing altitude from the 200 m mark. The microbial community in the water column above ANA site, consisted of over 50% SUP05 (composed of three ASVs: SUP05_1 and _2) and 19%





Sulfurovaceae (sulfur-oxidizing Epsilonproteobacteria family) from the seafloor to 250 m altitude. In contrast to Hatoma Knoll, methanotrophic lineages were not detected at ANA site.

## 4 Discussion

### 4.1 Active aerobic methanotrophy in Hatoma Knoll plume

Multiple lines of evidences point to the occurrence of microbial oxidation of $CH_4$ in the hydrothermal plume above Hatoma Knoll, between water column at the seafloor to 200 m altitude, but not at ANA site. Manganese concentration has been utilized as an indicator for the dilution of Mn-rich hydrothermal vent fluid with Mn-depleted ambient seawater, owing to the long life of manganese compared to particles and aerobically energetic molecules like $CH_4$ [Kadko et al., 1990]. In principle, a two-component mixing is represented by a straight line on the Mn plot, and a downward deviation from the ideal mixing line suggests significant removal of counterpart compound [German and Seyfried, 2014]. Indeed, the Mn plot of waters above the Hatoma Knoll exhibited downward convex curves and not a straight line for $CH_4$ concentrations, ammonium concentrations, and turbidity (Figure 6). The $NH_4$-Mn dataset of the Hatoma plume observed are shifted downwards from the ideal mixing line, assuming $NH_4$/Mn ratios of 11-18 observed in the Hatoma vent fluid [Toki et al., 2016], confirming the removal of ammonium from the plume. Despite the lack of available $CH_4$/Mn data for the estimated endmember fluid at Hatoma Knoll vent site [Toki et al., 2016], the downward convex curve of $CH_4$ concentrations on the Mn plot strongly suggests $CH_4$ removal. From the viewpoint of microbial ecology, the ATP-rich microbial cells (Figure 4) and the significant appearance of aerobic methanotrophic lineage (Figure 5) observed in the plume above Hatoma Knoll are both strongly indicative of active microbial methanotrophy within the plume. On the other hand, the invariability of $CH_4$/Mn ratios above ANA site regardless of concentrations suggests negligible $CH_4$ consumption in the plume. No dominant methanotrophic microbial groups being detected from the 16S rRNA gene community analysis above ANA site support a lack of methanotrophic activity.

The contrasting microbial community composition and methanotrophic activity seen between plumes above Hatoma Knoll and ANA sites are likely attributable to differences in the vent fluid chemistry. High-temperature hydrothermal fluids directly collected from vent orifices at Hatoma Knoll showed relatively high $CH_4$ concentrations sometimes >10 mM [Toki et al. 2016], in line with significant methanotrophic activity being detected in the Hatoma plume. In the plume originating from the $CH_4$-rich Guaymas Basin site [McDermott et al., 2015], the presence of Methylococcaceae and the high expression of methane oxidation gene (pmo) were revealed [Lesniewski et al., 2012]. In contrast, high-temperature vent fluid collected from ANA site only contained $CH_4$ at concentrations below 1 mM [Makabe et al., 2016], which explains the lack of significant methanotrophic activity in the plume above ANA site. The increase of total cell abundance (Figure 2) and the dominance of the plume-associated sulfur-oxidizing bacteria group SUP05 (Figure 5) [Sunamura et al., 2004] in the ANA



plume are evidences for significant shifts of the entire microbial community supported by hydrothermal fluids. Previously,
the plume above the Hakurei hydrothermal site in Izena Hole, Okinawa Trough was shown to exhibit anomalous $N_2O$
increase with 15N-depleted signature [Kawagucci et al., 2010] and raised the possibility in determining isotope fractionation
factors of $N_2O$ production in deep-sea water. The invariability of the $N_2O$ observed in this study, however, obscures
discussion about any isotopic effects with respect to $N_2O$.

### 4.2 Determination of isotope fractionation factors at Hatoma Knoll

Hereafter, we assume the $CH_4/Mn$ ratios of the plume above Hatoma vary only by aerobic $CH_4$ oxidation, for evaluating
isotope fractionation factors. When the $\delta^{13}C_{CH4}$ and $\delta D_{CH4}$ values of the Hatoma Knoll plume are plotted as a function of the
$CH_4/Mn$ ratio, there are observable increments of δ values along decrements of $CH_4/Mn$ ratio (Figure 7). This phenomenon
is well explained by the kinetic isotope effect on aerobic $CH_4$ oxidation which causes the enrichment of heavier
isotopologues, $^{13}CH_4$ and $CH_3D$, in the remnant reactant. Some large scatter of the $\delta^{13}C_{CH4}$ and $\delta D_{CH4}$ values are seen,
particularly at low $CH_4/Mn$ ranges (e.g. <10) – these data include two distinct seawater masses below 200 m altitude (the
height of the caldera wall), corresponding to the high pATP water inside the caldera (Figure 2) and waters coming in from
above 200 m (Figure 1). As our aim is to determine the isotope fractionation factors from the observed values, the gradual
changes in isotope composition within the water mass below 200 m altitude are used for further analysis.


According to previous studies [e.g. Gamo et al., 2010], we applied $CH_4/Mn$ ratio instead of f for the equation (2), as follows:

$$\varepsilon^a = [\delta_{rx}-\delta_{r0}] / \ln([CH_4/Mn]_{r0}-[CH_4/Mn]_{rx}) \quad (5),$$

where a represents carbon (C) or hydrogen (H) isotopes. The δ-ln[$CH_4/Mn$] plot analysis for the plume above Hatoma Knoll
yielded slopes representing $\varepsilon^C$ and $\varepsilon^H$ values of 5.2±0.4‰ and 49.4±5.0‰, respectively (Figure 7). The $\varepsilon^C$ and $\varepsilon^H$ values in
turn yielded a Λ value of 9.4 according to equation (3). This ε-based Λ value calculated from the plume sample is similar to
the δ-based Λ value calculated using the entire dataset from Hatoma Knoll (8.8)(Figure 3).

The $\varepsilon^H$ value of 49.4±5.0‰ determined is the first $\varepsilon^H$ value reported for aerobic $CH_4$ oxidation in the oxic seawater column.
The $\varepsilon^H$ value in seawater column occupied by Methylococcaceae (49.4±5.0‰) is comparable with those determined by
observations for terrestrial ecosystems (≥42‰) but clearly lower than those from aerobic and anaerobic methanotroph
enrichment cultures (93–320‰) [e.g., Rasigraf et al., 2012; Ono et al., 2021] as well as incubations of methanotrophic
isolates (110–232‰) [Feisthauer et al., 2011]. Remarkably, the methanotrophic isolate Methylococcus capsulatus, a
representative species of family Methylococcaceae, exhibited $\varepsilon^H$ values of 192‰ and 232‰ when cultivated at 45°C
with/without sufficient copper supply [Feisthauer et al., 2011].




The $\varepsilon^C$ values obtained through observations of deep-sea hydrothermal plumes reported to date are comparable among the sites and regions, including the 27.5°–32.5°S area on the East Pacific Rise (4–6‰)[Gharib et al., 2005], Myojin Knoll field on the Izu-Ogasawara Arc (5±1)[Tsunogai et al., 2000], Daiyon-Yonaguni site in the Okinawa Trough (5‰ and 12‰)[Gamo et al., 2010], Izena Hole also in the Okinawa Trough (<7‰)[Kawagucci et al., 2010], and Hatoma Knoll (5.2±0.4‰)[This

study]. These $\varepsilon^C$ values from the deep-sea plumes are lower than those estimated from methanotrophic isolate incubations (18.8–27.9‰) and methanotrophic communities (7.9–26.6‰) when not considering some exceptional values [e.g., Feisthauer et al., 2011]. Regardless of the $\varepsilon^H$ and $\varepsilon^C$ values, $\Lambda$ value of the hydrothermal plume (9.4 and 8.8) reported herein is consistent with those from methanotrophic isolate incubations (7.3–10.5), including the cultivation of M. capsulatus [Feisthauer et al., 2011]. The consistencies in both $\varepsilon^C$ values among deep-sea plumes and $\Lambda$ values among isolates and

plumes suggest that the $\varepsilon^H$, $\varepsilon^C$, and $\Lambda$ values determined in this study are appropriate. However, the reasons why the $\varepsilon^H$ and $\varepsilon^C$ values are smaller in deep-sea water columns compared to others remain unclear at this point. Lower temperatures and substrate concentrations of the deep-sea plume compared with the cultivations may contribute to the small fractionations. Future applications of the approach used herein to the other marine habitats with methanotrophic activity will reveal whether or not the small $\varepsilon^H$ value observed here is ubiquitous in the marine realm.


**5 Concluding Remarks**

The $\varepsilon^H$, $\varepsilon^C$, and $\Lambda$ values associated with aerobic $CH_4$ oxidation in seawater column above hydrothermally active areas determined herein are useful for understanding marine $CH_4$ dynamics. The $\delta D_{CH_4}$ values of seafloor hydrothermal vent fluids and hydrocarbon seep fluids are expected to be -130‰ and -180‰, respectively [Whiticar, 1986; Okumura et al., 2016]. As

such, observations of the $\delta D_{CH_4}$ values in the water column above the geofluid sites, combined with the $\varepsilon^H$ value of 49.4±5.0, enables us to estimate how $CH_4$ oxidation has progressed through the equation (2). The estimation of the fraction as well as the $\delta^{13}C_{CH_4}$ values of vent plumes further allow us to estimate the $\delta^{13}C_{CH_4}$ value of the endmember effluent $CH_4$. The estimated $\delta^{13}C_{CH_4}$ value of the endmember geofluid denotes the origin of $CH_4$ there, allowing 'sneak peeks' of the ongoing subseafloor process.


Our approach to determine isotope fractionation factors in seawater column environment by high-resolution hydrothermal plume sampling can be applied not only to conventional carbon and hydrogen isotope ratios of $CH_4$, but also 'clumped' isotope composition of $CH_4$ [Ono et al., 2021]. The same approach at Hatoma Knoll is also applicable to determining isotope fractionation factors for ammonium oxidation in marine environment because of the decreases of $NH_4$/Mn ratios (Figure 6)

and significant appearance of the ammonium-oxidizing archaeal family Nitrosopumilaceae (Figure 5). If the sulfur isotope fractionations associated with aerobic sulfide oxidation are interested, the same approach at ANA site would be appropriate because of microbial community composition occupied by sulfur-oxidizing members (Figure 5). Drastic changes in $N_2O$ and



$H_2$ concentrations in plumes at the Izena Cauldron [Kawagucci et al., 2010] also allow us to determine the isotope fractionation factors associated with their metabolisms in water column by the same approach. These are foci in future

studies.

**Data availability**

Dataset reported is available in Supplementary Table S1.


**Supplement file**

Supplementary Table S1: All analytical results drawn in figures.

**Author contribution**

SK and TY designed the study. YM, AM, TF, YO, TN, and TY conducted chemical and microbiological analyses. SK made a draft. All authors contributed to sampling and gave final approval for submission and publication.

**Competing interest**

The authors declare that they have no conflict of interest.


**Acknowledgement**

First of all, our biggest thanks go to Keiko Tanaka for her earnestness in the room Deep301. Manganese analyses were supported by Kaiyo Keisoku Co. Ltd. (president: Prof. Kei Okamura). The authors thank Dr. Hiroyuki Yamamoto, the master, crews, and scientific parties including MWJ and NME staffs of R/V Mirai cruise (MR17-03C) for their support. The

authors also thank Masami Koizumi, Miho Hirai, and Yoshihiro Takaki for assisting with microbiological analyses. Dr. Chong Chen proofread an earlier version of the manuscript to improve the English language. This study was supported by Council for Science, Technology, and Innovation (CSTI) as the Cross Ministerial Strategic Innovation Promotion Program (SIP), Next-generation Technology for Ocean Resource Exploration. This work was also supported by JSPS KAKENHI Grant Numbers 17H01869 and 20H02020. We thank All Nippon Airways (ANA) for providing a comfortable flight to

Okinawa Island, from where we embarked on the cruise to ANA site (and Hatoma Knoll).

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






**Figure 1: Seafloor topography of (a) Hatoma Knoll site and (b) ANA site.**
Sampling locations are indicated as cross points of dotted lines.

**Figure. 2: Vertical profiles of measured parameters.**
Y-axis represents altitude from the seafloor at each site. (a) Relative temperature from the seafloor value. (b) DO level ($\mu$mol kg$^{-1}$). (c) Turbidity (FTU). (d-e) Manganese and CH$_4$ concentrations (nmol L$^{-1}$). (f) Mehtane/Manganese ratio. (g-h) Carbon and hydrogen isotope ratios of CH$_4$ (‰). (i-j) Concentrations of ammonium (nmol kg$^{-1}$) and N$_2$O (nmol L$^{-1}$). (k-l) Nitrogen and oxygen isotope ratios of N$_2$O (‰). (m) Total cell density (coloured) as well as UCO (grey) with logarithmic x-axis (cell mL$^{-1}$). (n) pATP concentration (pmol L$^{-1}$)

**Figure. 3: $^{13}$C-D diagram for CH$_4$.**
Symbols of Hatoma Knoll samples are classified by color according to sampling altitudes below 200 m (blue) and above 200 m (pink). Grey diagonal line represents a linear fitting for the Hatoma knoll dataset.

**Figure. 4: A cross plot between pATP concentration and total cell density.**
Symbols are the same as those in Figure 3. Diagonal broken lines represent cellular ATP contents.

**Figure. 5: Prokaryotic composition of (a) Hatoma Knoll plume and (b) ANA plume.**
The top 10% (49 ASVs) of 4,880 ASVs, in terms of total reads abundance in this study, was defined as the major group. Read abundance of the major group corresponds to 83% of the total read abundance of the samples. The names and colours in the legend are identical in these two panels.

**Figure. 6: Manganese plots with CH$_4$ concentration, ammonium concentration, and turbidity.**
Symbols are the same as those in Figure 3. Grey triangle in panel (b) represents the ideal mixing line between ambient seawater and the 580 Hatoma vent fluid having NH4/Mn ratios of 11-18. Diagonal dot lines in panels (a) and (c) represent estimated mixing lines between ambient seawater and hydrothermal plume source, assumed by the highest CH$_4$/Mn and Turbidity /Mn ratios observed.

**Figure. 7: Isotope ratio changes along with aerobic methanotrophy above Hatoma Knoll.**
Panels (a-b) show CH$_4$/Mn ratios while panel (c-d) represent equation (5) in main text. Dash lines and grey zones illustrated in panels (c-d) represent fitting lines with standard errors corresponding to isotope fractionation factors of the $\varepsilon^C$ and $\varepsilon^H$ valeus.






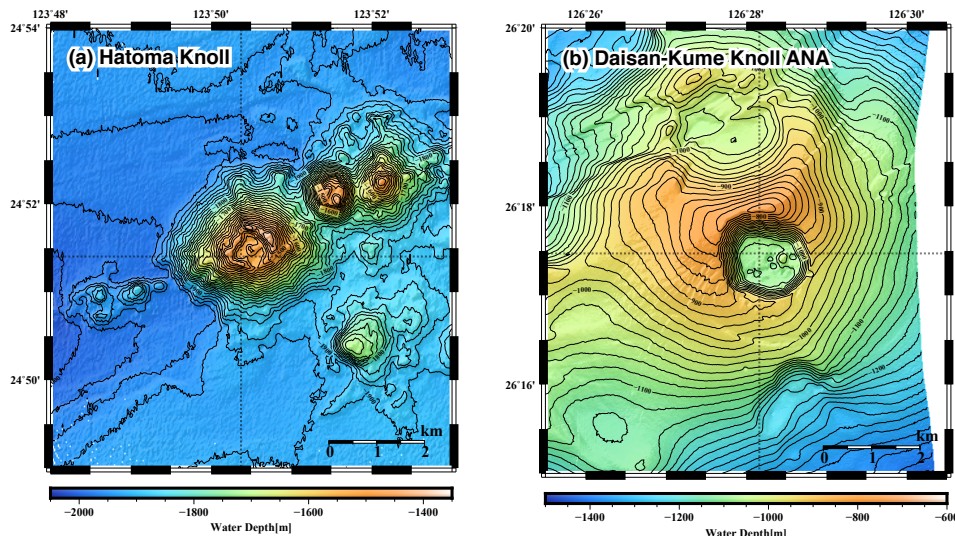

*Kawagucci et al. Figure 1*





*Kawagucci et al. Figure 2*



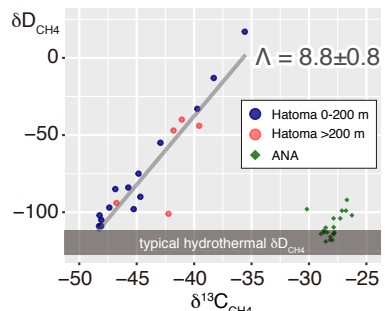

**Kawagucci et al. Figure 3**



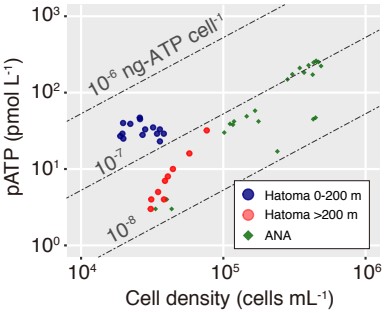

**Kawagucci et al. Figure 4**



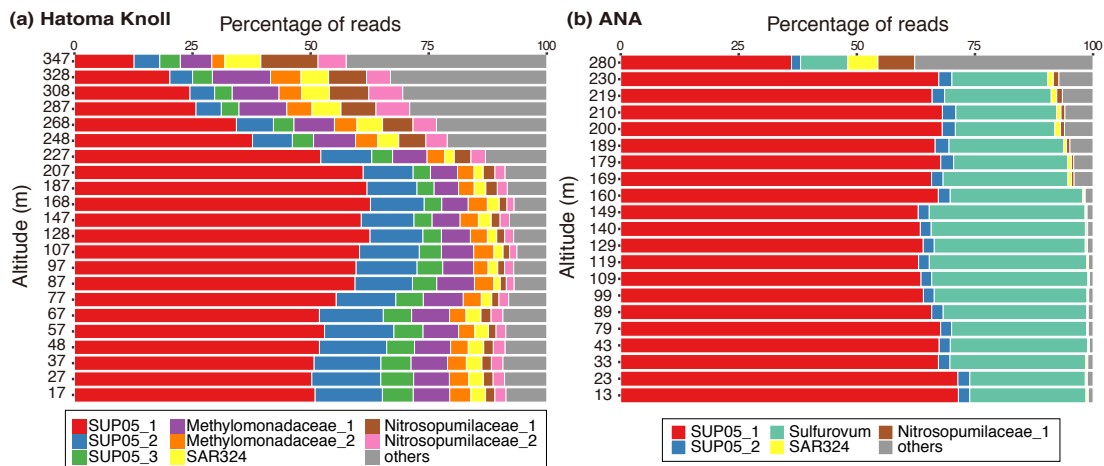

**Kawagucci et al. Figure 5**



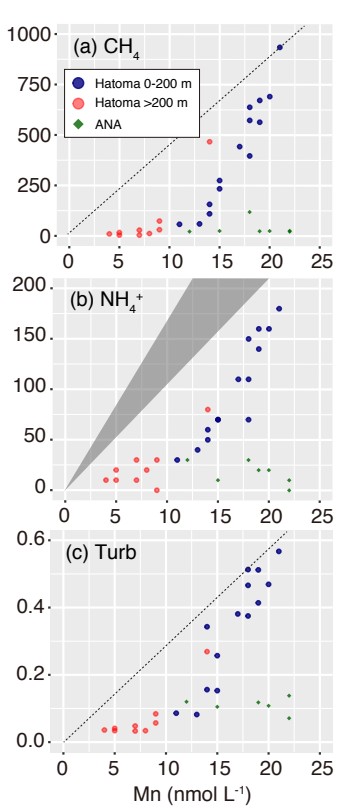

*Kawagucci et al. Figure 6*





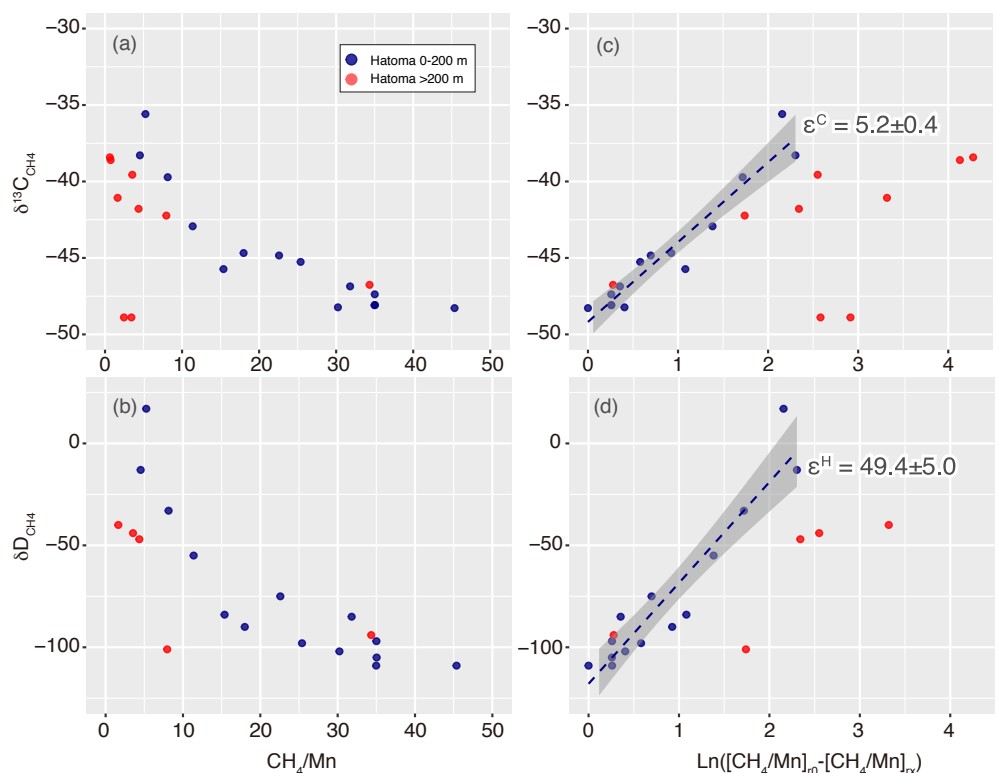

**Kawagucci et al. Figure 7**