# Peer review of "Hydrogen and carbon isotope fractionation factors of aerobic methane oxidation in deep-sea water"

_Biogeosciences, 2021_

## Author Response (AR1)

Carsten Vogt
General comments

The global methane cycle is in view of climate change an important biogeochemical topic, hence the study addresses relevant scientific questions within the scope of BG. The presented concept of two-dimensional stable isotope analysis of methane for describing methane removal processes is not new but has been applied for the first time in a marine water body, hence the data are novel and original and of relevance. The used scientific methods and assumptions are valid and clearly outlined. The reasons for analysing stable isotopes of N2O becomes not clear, however (see comments below). The data are sufficient to support the interpretation and conclusions. The reasons for lacking aerobic methane biodegradation in the ANA site are not clear however and might be discussed in more detail. The results are traceable, experiments, methods and calculations are described in detail and sufficient. The authors give overall proper credit to related work and indicate their own/ new contribution. The title reflect the content of the paper. The summary is concise and complete except the missing N2O stable isotope data, which should be either indicated in the abstract and explained in more detail in the main manuscript, or deleted. The overall presentation is well structured, the language is precise and fluent. Mathematical formulae, symbols, abbreviations, and units are correctly used. The N2O part of the paper should be clarified or eliminated. Number and quality of references is fine. The supplementary material is an excel sheet containing raw data which should be explained in more detail (e.g., units the for shown data are not given in the head of the respective data column) to facilitate the understanding of the data.

*---We appreciate the fair evaluation. For N2O, we decided to report only method in main text and the dataset moved to Supplementary Table. Supplementary Table was also modified according to the suggestion. Replies to each specific comment are below.*

Specific comments

L 37-38: Isotope fractionation factors are not always coupled to a 'series of multi-step enzyme-catalyzed reactions', this statement is slightly misleading. Indeed,

stable isotopes of methane analyzed to describe methanotrophic pathways are linked to single reactions catalysed by single enzymes (methane monooxygenase for aerobic methane oxidation or the first step of reversed methane oxidation for anaerobic methane oxidation). Actually, what I miss in the introduction is a brief description of the biochemical background of aerobic methane oxidation to clarify the mechanism of isotope fractionation upon methane oxidation.

*---We agree with the comment. We deleted the phrase 'series of multi-step enzyme-catalyzed reactions'. In addition, a brief description of the biochemical background of methane consumption was added to third paragraph as 'Kinetic isotope effect on cleavage of C-H bond of CH4 causes the isotope fractionation in the remnant CH4' and '... the microbial consumption of CH4, methanotrophy, mediated by enzymes such as methane monooxygenase...'.*

L 38: 'Fluctuate' is in my view not the precise word to describe changing isotope fractionation factors of a distinct (bio)chemical reaction due to changing conditions. Each reaction is characterized by a distinct isotope fractionation factor, which can be however **masked** due to abiotic, non-destructive 'dilution' effects.

*---We agree and disagree with the comment. Isotope fractionation is caused by kinetic isotope effect, which is the difference of reaction kinetics between each isotopologue. Reaction kinetics are strictly determined by given reaction conditions, as pointed out. On the other hand, reaction kinetics and the kinetic isotope effects are variable according to changes in reaction conditions such as temperature. We thus revised a word "fluctuate" to "variable".*

L 175: By using a 0.2 µM pore-size filter, ultra-small bacterial cells which may represent a substantial fraction of the total cell counts, will not be counted. This should be discussed.

*---It may be a misreading. For cell count, we used unfiltered sample as stated in L169-174 of the original manuscript.*

L 196-197: Temperature units are not given in Figure 2,

Figure 2: Units for temperature are between 0°C and 0.3°C? Unclear.

> *---Temperature profiles in Figure 2 were now shown in seawater temperature (°C).*

Figure 3: Why no Λ value for the isotope data of the ANA site is given?

> *---Little consumption of methane and insignificant changes of both d13C and dD do not allow appropriate evaluation of Λ value, although it looks available for Λ calculation.*

L 290-292: Lower but measurable (and constantly available) concentrations of methane should allow methanotrophs to grow, I do not understand the argumentation here. Please explain.

> *---We added the sentences for clarification of this issue. 'Although available CH4 likely fuels the methanotrophs, the low concentration allows only slow methanotrophic activity compared to the residence time of the plume. The slow rate could result in undetectable signature of the methane consumption in concentrations and isotope ratios if the available CH4 remains in water column..'*

L 292-294: the 16S rRNA gene data indicate that inorganic sulfur compounds are the main electron donors in both investigated systems. It would be excellent if the authors could provide additional data on, e.g. concentrations of inorganic sulfur compounds and integrate them into Figure 2, to support this hypothesis

> *---We agree with the comment that sulfur metabolisms in the plume should be investigated. However, we did not collect/analyze any sulfur compound in the cruise. That will be done in near future.*

L 294-299: I wonder why the results about $N_2O$ stable isotopes were not mentioned in the abstract, since the data seem to be exceptional. In this context, I also wonder why the $N_2O$ topic has not been briefly described in the introduction. The background and goal of the $N_2O$ stable isotope analyses becomes not clear. Thus, I suggest either deleting these data to streamline the methane story, or to integrate the $N_2O$ story into the manuscript by explaining the aims of this study in more detail.

   *---See reply to general comment.*

L 335-339: The lower absolute fractionation for carbon and hydrogen upon aerobic methane oxidation point to a considerable masking of isotope fractionation in the water column, e.g. due to limited mass-transfer of methane to the methane monooxygenase inside the cells, or a considerable decrease of methane concentrations in the water column due to abiotic, non-fractionating processes (e.g., dispersion, dilution). Notably, similar differences in absolute isotope fractionation for carbon and hydrogen were observed for anaerobic benzene degradation at laboratory and field scale by Fischer et al. (2009) Rapid Communications in Mass Spectrometry 23: 2439-2447, this study might be discussed here for comparison.

   *---We appreciate the insightful comment and the introduction of reference. In our case, where CH/4Mn ratio is used instead of [CH4] to eliminate factors of dilution/dispersion, the small isotope fractionation factor can be derived from 'limited mass-transfer of methane'. The sentences were revised to "A possible explanation, originally proposed for the case of benzene biodegradation [Fischer et al., 2009], is that the cells take up only a limited amount of methane which is then virtually all consumed, leading to little change in the isotopic ratios of water column methane.".*

L 20-21: 'Although the isotope fractionation factors associated with methanotrophy been examined under various conditions, ….' – have been examined

L 566: Mehtane – change to Methane

   *---Revised, thank you.*

Jeff Chanton

I was asked to review this paper by the editor Jack Middelburg. I agree with the authors treatment of the data, and their interprtation of it.

1. I do think it is important that they clearly state the temperature of the seawater where they made their measurement. They state that the vent flued temp was 229C in line 91 and the other site was 323C, line 88. In the graphs, figure 2, they report the temperature differential. Is that relative to these reported fluid tems? or to what.?? The fraction factor for methane oxidation is sensitive to temperature as found in the reference below, so the auhtors should be crystal clear about the temperature at whech they made their measurements.
Chanton, J. P., D. K. Powelson, T. Abichou, D. Fields, & R. B. Green. 2008. Effect of Temperature and Oxidation Rate on Carbon-isotope Fractionation during Methane Oxidation by Landfill Cover Materials, Environmental Science and Technology No 42, pp 7818-7823. DOI 10.1021/es80122y.

*---We appreciate the comments. Water temperatures were described as 'temperature were drawn as difference from the bottom temperature (Figure 2a), which was 3.79 at Hatoma Knoll and 4.45 at ANA site' at L104-105 of original manuscript. We believe that the difference of these temperatures is negligible in terms of microbial physiology. By the way, for more clear presentation, we revised the temperature profiles in Figure 2 based on temperature, not the temperature difference from the bottom. In addition, we added a sentence 'temperatures of seawater collected ranged between 3.5°C and 7.0°C (Figure 2a)' in Results chapter.*

2. Rather than call the height of the water column above the seafloor as altitude, it should be refered to as heifht above the sea floor.

*---We know both altitude and height are used in community studying the hydrothermal systems. Here we decide to use altitude, not height. Thanks.*

---

## Editor Decision (ED1)

[revised manuscript text omitted]

*Kawagucci et al. Figure 1*

[Figure]

*Kawagucci et al. Figure 2*

[Figure]

**Kawagucci et al. Figure 3**

[Figure]

**Kawagucci et al. Figure 4**

[Figure]

**(a) Hatoma Knoll**

Percentage of reads

Altitude (m)

Legend:
- SUP05_1
- SUP05_2
- SUP05_3
- Methylomonadaceae_1
- Methylomonadaceae_2
- SAR324
- Nitrosopumilaceae_1
- Nitrosopumilaceae_2
- others

**(b) ANA**

Percentage of reads

Altitude (m)

Legend:
- SUP05_1
- SUP05_2
- Sulfurovum
- SAR324
- Nitrosopumilaceae_1
- others

*Kawagucci et al. Figure 5*

[Figure]

*Kawagucci et al. Figure 6*

[Figure]

**Kawagucci et al. Figure 7**